# Selection for Growth Performance in *Oreochromis niloticus* Across Different Aquatic Environments Using Growth Hormone Gene Polymorphisms

**DOI:** 10.3390/ani15142097

**Published:** 2025-07-16

**Authors:** Fawzia S. Ali, Simone T. Awad, Mohamed Ismail, Shabaan A. Hemeda, Abeer F. El Nahas, Eman M. Abbas, Ahmed Mamoon, Hani Nazmi, Ehab El-Haroun

**Affiliations:** 1Aquaculture Division, National Institute of Oceanography and Fisheries, NIOF, Cairo 11516, Egypt; fawzias.ali84@gmail.com (F.S.A.); simona158@yahoo.com (S.T.A.); emanabbas03@yahoo.com (E.M.A.); hani_nazmi@hotmail.com (H.N.); 2Department of Animal Husbandry and Animal Wealth Development, Faculty of Veterinary Medicine, Alexandria University, Alexandria 22758, Egypt; shabaan.hemeda@yahoo.com; 3Genetics Department, Faculty of Agriculture, Menoufia University, Shibin El-Kom 6131567, Egypt; mohamed.ismail@agr.menofia.edu.eg; 4Fish Production Department, Faculty of Agriculture, Al-Azhar University, Cairo 11884, Egypt; ahmed.mamoon@azhar.edu.eg; 5Department of Integrative Agriculture, College of Agriculture and Veterinary Medicine, United Arab Emirates University, Al Ain P.O. Box 15551, United Arab Emirates

**Keywords:** Nile tilapia (*Oreochromis niloticus*), selective breeding, growth hormone gene, SNPs, population structure, larval gene expression

## Abstract

Nile tilapia is a widely farmed fish species in tropical aquaculture due to its high growth, adaptability, and palatability. The growing global demand for sustainable protein sources prioritizes the importance of enhancing the growth performance of Nile tilapia through selective breeding. This study focused on the naturally occurring genetic differences linked to the growth hormone gene and their association with body weight and length in three Egyptian wild Nile tilapia populations (Lake Brullus, El-Qanater El-Khairia, and Lake Nasser). Populations from Lake Nasser and El-Qanater El-Khairia showed the highest numbers of beneficial genetic markers and elevated body measurements. Furthermore, their offspring demonstrated improved early growth, immunity, and neural performance. These findings clarify the importance of using genetic information in Nile tilapia breeding strategies. This approach guarantees the long-term sustainability of aquaculture, particularly in developing countries where fish farming plays a critical role in food security and economic development.

## 1. Introduction

Aquaculture represents a promising solution for fulfilling the burgeoning demand for nutritious, affordable, and readily available animal protein [1,2]. The larval stage is considered the weakest link in aquaculture due to its elevated sensitivity and nutritional requirements, in addition to the undeveloped immune response [3,4]. Egypt is considered one of the cornerstones in the tropical aquaculture industry due to its ideal climatic conditions and proximity to the Nile River, which positively affect its economy and food security [5]. Selective breeding programs are essential for improving aquaculture and will result in production surpassing the limit of 100 million tons by 2027 [1]. This approach aims to solve enduring challenges in the field through the careful selection and breeding of individuals with superior traits, for instance, an accelerated growth rate, enhanced disease resistance and survival, elevated feed conversion efficiency, and high offspring quality (viability and resilience) [2,6,7,8,9].

Traditionally, selective breeding was based on visual phenotypic evaluation to select individuals with favorable traits as broodstocks for subsequent generations [10]. However, this technique faced several limitations, including bias, an elongated temporal span, and incompetence in polygenic trait selection [11]. Therefore, the shift toward marker-assisted selection (MAS) has emerged as a revolutionary approach to uncover the precise DNA sequences related to superior traits [12,13].

Polymorphisms in growth-related genes, including insulin-like growth factor I (*IGFI*), somatostatin (*SST*), insulin-like growth factor II (*IGFII*), and growth hormone (*GH*)*,* are linked to variations in growth performance in fish. Polymorphisms within the promoter and/or exon regions create changes within the binding sites of transcription factors, significantly affecting gene expression [14,15,16,17,18]. *GH* is the cornerstone of the hypothalamic–pituitary–somatotropic (HPS) axis and affects a wide range of physiological activities related to somatic growth, including energy utilization, muscular development (hyperplasia and hypertrophy), food conversion efficiency, and regulation of sexual maturation and immunity [16,17,18,19].

MAS mainly utilizes different DNA markers in aquaculture, including simple sequence repeats (SSRs) and single-nucleotide polymorphisms (SNPs) [20,21,22,23]. Such DNA markers enable the identification of genetic variations linked to enhanced genetic performance [20,24]. SNPs are mutation-induced nucleotide substitutions, distributed throughout the genome (coding and noncoding regions); they can serve as biological markers, assisting in identifying genes linked to productive traits [25,26]. However, despite extensive use of MAS in aquaculture, there remains a limited understanding of how specific SNPs in growth-related genes translate into phenotypic and molecular outcomes in early progeny performance [18].

The Nile tilapia, *Oreochromis niloticus*, is a highly valued species in tropical aquaculture because of its resilience, rapid growth rate, high reproducibility, and widespread consumer acceptability [27]. Therefore, it has become a candidate for several genetic improvement programs worldwide, which resulted in producing the first genetically improved farmed tilapia (GIFT) program established by WorldFish in 1988 until the latest generation of the Genomar Supreme Tilapia (GST) program [1,17,18,28,29]. To date, the genetic pool of Egyptian Nile tilapia has contributed to the creation of eight genetically improved strains, which are now cultivated in more than 128 countries worldwide [30,31].

Although Egypt has made a substantial contribution to the global tilapia aquaculture industry, the genetic potential of its native Nile tilapia populations is still being underutilised. Hence, these populations represent hidden reservoirs of natural genetic diversity, particularly in geographically distinct aquatic ecosystems, which might house unique alleles associated with superior growth and resilience characteristics [27].

The current work represents a part of a currently running genetic improvement program for Nile tilapia, *Oreochromis niloticus*, at the National Institute of Oceanography and Fisheries, Egypt. This program is based on the screening and selection of wild Nile tilapia, *Oreochromis niloticus*, populations from three dispersed geographic locations, including Lake Brullus (B) in north Egypt, El-Qanater El-Khairia (Q) an intermediate Egyptian locality at the Nile River, and Lake Nasser (A) in south Egypt, for subsequent hybridizations. The objectives of this study were to investigate the presence of SNPs within the *Oreochromis niloticus* growth hormone gene, to conduct marker-trait association between the identified SNPs and growth performance (body weight and length), to select individuals with superior growth performance bearing such beneficial SNPs as broad stocks for the subsequent hybridization process, and to evaluate the early offspring performance in 72-h-old larvae obtained from selected broadstocks.

## 2. Materials and Methods

### 2.1. Ethical Statement

The relevant international and institutional guidelines pertaining to the moral care and application of animals have been adhered to. The National Institute of Oceanography and Fisheries (NIOF) Committee for Ethical Care and Use of Animals/Aquatic Animals (NIOF-IACUC) extensively reviewed the research protocol and assigned an acceptance code (NIOF-AQ6F22P031).

### 2.2. Broadstock Collection, Adaptation, and Tagging

Nile tilapia, *Oreochromis niloticus*, wild populations were obtained from three Egyptian water bodies: Lake Brullus (B), the River Nile at El-Qanater El-Khairia (Q), and Lake Nasser (A) (Figure 1).

A total of 155 fish samples (89 females and 66 males) of the same age (one year old; ageing was conducted by direct scale examination [33]), with lengths ranging from 17.5 to 24 cm and weights ranging from 252 to 470 gm, were collected from the three locations. The fish stocks were transferred in aerated fish tanks to the National Institute of Oceanography and Fisheries (NIOF), El-Qanater El-Khairia, Egypt. The fish were then sorted according to their sex in separate fibreglass tanks (2 m^3^) with a suitable density (5.3 kg/m^3^), constituting six groups (males EL-Qanater (MQ: 23), males Lake Brullus (MB: 22), males Lake Nasser (MA: 21), females EL-Qanater (FQ: 15), females Lake Brullus (FB: 31), and females Lake Nasser (FA: 43)), with sex ratios (male: female) of 1.53:1, 0.71:1, and 0.49:1 in the EL-Qanater, Lake Brullus, and Lake Nasser populations, respectively. The fish stocks were kept in quarantine for 30 days to neutralize the influence of prior environmental exposures and synchronize metabolic baselines, during which any fish with abnormal physical activity and/or feeding response were excluded [34,35,36].

Fish samples were labelled with a TG2-F tagging gun (Dynamic Aqua Supply Ltd., Canada) (Dynamic Aqua Supply Ltd., Surrey, BC, Canada). The weight and length of all individuals were recorded monthly. Each group was reared in separate tanks with an average stocking density of 5.3 kg/m^3^, and the feeding regime was continuously adjusted according to the standing stock biomass (3%) and crude protein % (30.3) [37].

The water quality measurements, including temperature, pH, dissolved oxygen, and ammonia, were assessed daily throughout the experiment via a mercury thermometer, pH meter (Digital Mini-pH Meter, model 55, Fisher Scientific, Waltham, MA, USA), digital oximeter (Yellow Spring Instrument Co., model 58, Yellow Springs, OH, USA), and DREL/2 HACH kits (HACH Co., Loveland, CO, USA), and the results were verified as per the permissible limits set by the American Public Health Association. The average temperature, pH, DO, and NH_3_ were 28 °C, 7.3, >5.0 mg/L, and <0.02 mg/L, respectively; the daily water exchange rate was 20% of the tank’s capacity [38].

### 2.3. Parent Crosses and Fry Collection

The selected broodstocks were transferred into spawning tanks (2 m^3^) with a sex ratio of 1 male to 3 females (12 fish per tank: 3 males and 9 females). Spawning was allowed under semi-natural conditions without hormonal induction. Broodstock were fed a high-protein diet (38–40% crude protein) to support optimal reproductive performance and egg quality [39,40].

Fry were gently collected from the mouths of brooding females at 72 h post-hatch for gene expression analysis to mimic the natural maternal care of Nile tilapia species until sampling, while the 72 h time point represents a critical developmental phase where growth-related gene expression is activated as the fry approach exogenous feeding onset [41,42]

### 2.4. Genetic Analysis of the Broadstocks

#### 2.4.1. Broadstock’s DNA Extraction, PCR Primer Design, and Partial *GH* Gene Amplification and Sequencing

To ensure noninvasive DNA extraction, a 0.5 cm^2^ clip of the caudal fins of the tagged fish was collected via sharp, disinfected scissors and kept in absolute ethanol at −20 °C. Total genomic DNA was extracted from the tagged fish samples, including males and females (males EL-Qanater (MQ: 23), males Lake Brullus (MB: 22), males Lake Nasser (MA: 21), females EL-Qanater (FQ: 15), females Lake Brullus (FB: 31), and females Lake Nasser (FA: 43)), as described by Asahida et al. [43]. Specific primers for the growth hormone gene encoding *GH* were designed to partially amplify the Oreochromis niloticus growth hormone (*GH*) gene (GenBank accession No. M97766.1). Forward and reverse primers were designed using Primer3 [44]. The selected primer combinations were (forward, *GH*-1F: 5′-CGAGAAGGGACCACTGCTTT-3′) and (reverse, *GH*-1R: 5′-AGAGCTCTCCTATGACATGTGT-3′), targeting a fragment of 600 bp. Polymerase chain reaction (PCR) amplification was performed via an Applied Biosystems Verity 96-well thermocycler (Applied Biosystems, Foster City, CA, USA). The reaction mixture, with a total volume of 20 µL, composed of 12.5 µL of BIOLINE 2X My Taq Red Mix master mix, 1 µL of DNA template (approximately 20 ng), 1 µL of forward and reverse primers, and 4.5 µL of DNase/RNase free water using the following PCR conditions: initial denaturation at 94 °C for 4 min, followed by 35 amplification cycles consisting of denaturation for 1 min at 94 °C, annealing for 1 min at 62 °C, and a 90 s extension at 72 °C. The final step included a 10 min extension at 72 °C. The produced amplicons were purified as described by Megahed et al. [45] and validated by agarose gel electrophoresis (2%) before use in direct sequencing. The PCR products were sequenced using the same primers by Macrogen (Seoul, Republic of Korea).

#### 2.4.2. Sequence Analysis and SNP Discovery

To ensure high data accuracy and reliability, quality control (QC) of raw sequence data was carried out prior to alignment and SNP discovery. Chromas software (version 2.6.6) was initially used to visualize chromatogram files to evaluate base-calling accuracy and remove low-quality ends. The raw reads were evaluated using Phred quality scores calculated as Q = −10 × log_10_(*P*), where Q is the Phred quality score and P is the probability of an incorrect base call. Reads were trimmed to remove low-quality bases (Phred score < 20). Sequences with more than 5% ambiguous bases (N) or shorter than 100 bp after trimming were excluded from further analysis [46].

Sequence alignment and editing were conducted using BioEdit version 7.0.9.0 [47]; subsequently, the sequences were examined with the Basic Local Alignment Search Tool (BLAST) version 2.14.1 from the NCBI database to verify the accurate amplification of the target fragment of the *GH* gene. Sequence data produced in the current study have been deposited in the GenBank nucleotide sequence database (accession numbers LC832463–LC832617).

Single-nucleotide polymorphism (SNP) discovery was conducted via the online tool Sniplay3 [48]. Raw sequence data were pre-processed using vcftools v0.1.16 [49] as implemented in Sniplay3 with the following criteria to retain only high-confidence SNPs: (minQ 30; remove SNPs with quality scores below 30, minDP 10; ensures a minimum read depth of 10 per SNP, maxDP 100; ensures a maximum read depth of 100, maf 0.01; minor allele frequency ≥0.01, max-missing 0.90, retain SNPs in most samples).

Aligned FASTA files of the partial *GH* sequences accession numbers LC832463 to LC832617 were used to generate a consensus sequence, highlighting the most abundant nucleotides and their respective positions. This consensus sequence was then compared to each individual sequence to identify SNPs and their respective frequencies.

#### 2.4.3. Population Structure and Kinship Analysis Using SNP Data

Since the presence of population structure and/or kinship could affect genetic association outcomes, genotypic data from SNPs were used to determine the number of homogeneous populations using STRUCTURE software version 2.3.4 [50], applying a blind evaluation without previous population information and an admixture model with correlated allele frequencies. The number of clusters (*K*) was set between one and eight; runs had a burn-in period of 10^6^ iterations, followed by 10^6^ data collection iterations, and were repeated at least six times. The number of clusters was determined using the ad hoc statistic ∆*K*, and the assignment probabilities were visualized via DISTRUCT 1.1 [51]. Pairwise kinship coefficients among the studied genotypes were determined using the kinship procedure of Ritland [52] in SPAGeDi version 1.5 [53].

#### 2.4.4. Association Analysis Between SNPs and Body Characteristics

To determine the genetic correlation between the detected SNPs and body characteristics, the mixed linear model approach (MLM) was used while accounting for both fixed and random effects as implemented in TASSEL version 5.0 [54]. The model was defined as follows:*y* = X*β* + Q*γ* + Z*u* + *ϵ*
where *y* represents the observed phenotypic values. The fixed effect X*β* included population, location, age, initial weight, sex, and SNP genotype. The term Q*γ* accounts for the fixed effect of population structure. The random effect Z*u* represents kinship among individuals, where *u* and *ϵ* denote the residual error. This model allowed for accurate estimation of SNP effects while controlling for confounding effects due to relatedness and environmental variables. The inclusion of both population structure and kinship in MLM minimizes Type I error by leveraging relatedness and population structure effects.

False-positive results owing to population structure or familial relationships among the genotypes would be reduced if the association model included both elements [55]. Positive associations were detected at the *p* < 0.05 level, and all results were further corrected for multiple testing via Benjamini–Hochberg false discovery rate (FDR) procedure [56]. This approach ranks all observed *p*-values and adjusts them based on their rank and the total number of tests performed to control the expected proportion of false positives. SNPs with an FDR-adjusted *p*-value (q-value) below 0.05 were statistically significant. Applying the FDR allows the identification of true associations while minimizing false-positive associations due to multiple testing. The FDR thresholds were calculated using Microsoft Excel.

#### 2.4.5. Linkage Disequilibrium Analysis

For further detection of the candidate region(s) related to the two traits, a linkage disequilibrium analysis was conducted within the chromosomal regions using a sliding window of 300 bp using Haploview v4.2 [57]. The linkage disequilibrium analysis defined a block based on the solid spin algorithm according to the criteria of Gabriel et al. [58].

### 2.5. Isolation of RNA and cDNA Synthesis from 72-Hour Larvae

#### 2.5.1. RNA Extraction

RNA was extracted from selected broadstocks’ larvae 72 h post-hatch (three/group) and compared against the control group (three/group) of the same-aged larvae, which was obtained from a commercial hatchery located at Kafr El-Sheikh using ABTizol (Applied Biotechnology, Ismailia, Egypt) following the manufacturer’s guidelines. The purity and concentration of the extracted RNA were evaluated using NanoDrop at 260/280 nm (BioDrop, Cambridge, UK). The isolated RNA was utilized in cDNA synthesis using the ABT 2X RT mix (Applied Biotechnology, Ismailia, Egypt), complying with the producer’s directives. The cDNA was deep-frozen at −20 °C until further analysis. The cDNA was validated using polymerase chain reaction (PCR) using the reference gene (*βeta-actin*). Subsequently, the amplicon was examined on a 2% agarose gel electrophoresis and visualized using the gel documentation system (Biometra, Analytic Jena Company, Göttingen, Germany).

#### 2.5.2. Quantitative Real-Time PCR

The qRT-PCR was performed to analyse the RNA transcript of growth-related genes (growth hormone (*GH*) and insulin growth factor I (*IGFI*)), inflammatory-related genes (interleukin 1 beta (*IL1β*), interleukin 8 (*IL8*), CC chemokine (*CC*), CXC_2_ chemokine (*CXC_2_*), and tumor necrosis factor alpha (*TNFα*)), the immune-related gene (recombination activating gene (*Rag*)), and the neurological development-related gene (sacsin molecular chaperone (*Sacs*)) (Table 1).

The reaction mixture is composed of 10 μL SYBR green with low Rox (ABT 2X qPCR Mix (Applied Biotechnology, Egypt), 0.8 μL from each primer (R and F), 2 μL complementary DNA, and 6.4 μL of RNase-free water (total volume 20 μL). The thermal profile commences with an initial heating phase for 3 min at 95 °C, followed by 45 cycles, 15 s each at 95 °C, and an annealing for 1 min at 60 °C. The reaction was repeated twice. To test the precision of the PCR amplicons, the dissociation curve was made at the completion of the last cycle. This required collecting fluorescence data at 60 °C and checking every 7 s until the temperature reached 95 °C [32].

### 2.6. Statistical Analysis

Statistical analyses were performed using GraphPad Prism v8.1 (GraphPad Software, San Diego, CA, USA). For body weight and length (one-year-old fish specimens), a two-way analysis of variance (ANOVA) was conducted to evaluate the effects of sex and location (both treated as fixed effects). Where significant interactions were detected, Tukey’s post hoc test was used for pairwise comparisons.

The expression profiles of 72-hour-old larvae were analysed utilising comparative threshold cycle (CT). The results were recorded as fold change and compared to the control after normalisation by the *β-actin* gene as a reference gene [66,67,68].

Relative quantification of targeted genes was calculated using the comparative threshold cycle (CT) method following Rao et al. [69].

Where fold change = 2^−ΔΔct^.

Variations in gene expression results were interpreted by one-way ANOVA. GraphPad Prism software v8.1 (GraphPad Software, San Diego, CA, USA). The results were expressed as mean ± SE; significance was located at *p* ≤ 0.05. While tank effects were minimized by maintaining standardized rearing conditions and consistent density across tanks, all replicates were randomly distributed, and no significant tank-dependent variation was observed. Thus, random tank effects were not modeled statistically.

## 3. Results

### 3.1. Variation in Body Measurements Across Locations

Significant differences were observed in body weight and length among the three studied populations of *Oreochromis niloticus* (Lake Brullus, El-Qanater, and Lake Nasser). Moreover, within-population sex differences were evident, particularly in the River Nile (El-Qanater) and Lake Brullus populations. Notably, fish from Lake Nasser, both males and females, exhibited significantly higher morphometric values than their counterparts from the other two regions (Figure 2A,B).

### 3.2. Sequence Analysis and SNPs Discovery of the Growth Hormone (GH) Gene

Successful amplification of the targeted 600 bp region within the *GH* gene (covering parts of exons and introns) was achieved in all 155 tagged individuals. BLAST alignment confirmed specificity to the *GH* gene (GenBank accession no. M97766.1). A 585 bp consensus sequence was generated, and a total of 40 SNPs were initially identified through variant calling. Following quality control filtering criteria, which included thresholds for minimum quality score (QUAL ≥ 30), minimum read depth (DP ≥ 10), minor allele frequency (MAF ≥ 0.01), and a maximum of 10% missing data per SNP, fourteen high-quality SNPs passed all QC criteria (Figure 3). The selected SNPs were at detectable allele frequencies (i.e., MAF ≥ 0.01), ensuring adequate coverage across the dataset. The selected SNPs exhibited low missing genotypes (MAX of 10% missing data per SNP). These filtered variants were subsequently retained for downstream analyses.

### 3.3. Population Structure Assessment

STRUCTURE analysis based on *GH* gene SNPs identified two primary genetic clusters (*K* = 2) (Figure 4). However, this classification did not clearly separate the six tested subpopulations. Male groups from all three locations (MA, MQ, MB) exhibited relative genetic homogeneity, whereas female groups showed more variability.

### 3.4. Association Analysis Using SNPs

#### 3.4.1. Associations Between SNPs and Morphometric Traits

The results of significance testing for the fixed effects included in the MLM association model, including population, location, age, initial weight, sex, and SNP genotype, along with the corresponding *F*-statistics, *p*-values, and *R*^2^ values, are presented in Appendix A. Nine SNPs and three InDels were significantly associated with body weight and/or length, confirmed after false discovery rate (FDR) correction (Table 2).

The proportion of phenotypic variance (*R*^2^) explained by individual markers ranged from 2.6% (e.g., SNP3, SNP4 with body weight) to 36% (InDel3 with body weight). Several markers (SNP1, SNP8, InDel2, InDel3) were associated with both traits simultaneously (Figure 5).

#### 3.4.2. Linkage Disequilibrium Analysis

Three haplotype blocks were identified among the associated SNPs: Block 1 (SNP1, SNP2), Block 2 (SNP3–SNP6), and Block 3 (SNP7–SNP9), demonstrating strong linkage disequilibrium (Figure 6).

#### 3.4.3. Population-Specific Effects of SNPs

The effect of SNPs on morphometric traits varied among populations (Figure 7). All markers showed a general positive effect on body length across populations. MQ and MB exhibited the highest number of SNPs/InDels linked to increased body length. In contrast, MA showed unique SNPs (SNP3, SNP4, SNP9) and InDel3 associated with significantly higher body weight (Figure 7A,B).

#### 3.4.4. Genotypic Distribution of SNP-Associated Traits

The number of genotypes carrying the SNP-associated trait differed across populations (Table 3). MA (n = 39) and MQ (n = 38) had the highest genotype counts with significant SNP-trait correlations. In contrast, lower genotype counts were found in FB (n = 9) and FQ (n = 11). Genotype-SNP combinations associated with either or both body weight and length showed population-specific patterns.

### 3.5. Gene Expression Profiles in F_1_ Progeny

Significant upregulation of growth-related genes (*GH, IGFI*) was observed in F1 progeny of selected broodstock, particularly in the Lake Nasser group (*p* < 0.05), followed by El-Qanater and Lake Brullus (Figure 8). Proinflammatory cytokines *(IL1β, IL8*) and chemokines (*CXC*_2_) were also elevated in all tested progenies compared to the hatchery control group (*p* < 0.05). Notably, *TNFα* expression was significantly downregulated in Lake Nasser progeny compared to Lake Brullus (Figure 9). Moreover, genes associated with neural and immune development (*sacs*, *Rag*) were significantly upregulated in all groups except for Rag in Lake Brullus progeny, suggesting early ontogenetic activation of immune and neurodevelopmental pathways (Figure 10).

## 4. Discussion

Choosing the appropriate strain and implementing efficient feeding and rearing strategies are key components of successful aquaculture, as they assist in achieving the production objectives and reducing the input costs [70]. In aquaculture, growth performance is the most critical asset that should be enhanced [14,24]. Prior to conducting any genetic studies of the present research, multiple approaches were implemented to mitigate the ecological carryover effect that arose from the sampling site differences of the wild populations of Nile tilapia fish, including subjecting all fish samples to acclimatisation periods to help synchronise the metabolic baselines and using the monthly body measurement in the marker trait association analysis to account for individual growth dynamics over time; however, these inherited effects cannot be completely eliminated. Hence, this phenomenon is largely related to the complex interaction between environmental history, parental conditioning, and epigenetic influences, which can persist across generations even under controlled experimental circumstances [71,72].

In the current study, a partial sequence of the *GH* gene was analysed for the presence of SNPs, focusing on their potential application in association with growth performance traits. This approach enables the use of *GH* polymorphisms as a preliminary step in a genetic improvement program for Nile tilapia. *GH* performs a key role in several physiological processes, including muscle growth, efficient feed conversion, reproduction, immunity, salinity adaptation, and pollution resistance [16,73].

In the present study, the body measurements of Nile tilapia, *O. niloticus*, populations revealed significant differences across the three studied locations (Lake Brullus, EL-Qanater, and Lake Nasser). Compared to those of the EL-Qanater and Lake Brullus populations, the Lake Nasser population showed the highest body weight and length. The superior growth performance of Lake Nasser’s strains was reported previously. For instance, Awad et al. [32] reported the upregulated expression of the growth-linked genes in the *Oreochromis niloticus* population in Lake Nasser and the diminished heavy metal concentration in its water. Also, Abdel-tawwab et al. [74] reported the superior growth performance of Lake Nasser *Oreochromis niloticus* populations and attributed this growth performance to the interaction between genotype and the environment. On the other hand, Musa et al. [70] attributed the superior growth performance of this particular *Oreochromis niloticus* population to genetic factors. The genetic makeup of Lake Nasser’s strains, in addition to the exceptional water quality of the lake, may have synergistically contributed to the superior growth performance of this population [75,76].

Substantial variations in length were observed between males and females from Lake Brullus and El-Qanater; these differences might be associated with the divergent impacts of sex hormones on the growth performance of teleost fish [77,78]. Notably, the principal female sex hormone, oestrogen, reduces the responsiveness of hepatocytes to *GH* and suppresses the expression of the *IGFI* gene. Conversely, androgens, which are male hormones, increase the sensitivity of liver cells to *GH* and boost the expression of the *IGFI* gene, leading to enhanced growth performance [77,79].

Many factors affect the presence of SNPs, including endogenous elements like genetic tendency and reactive oxygen generated during regular cellular metabolism. In addition to mutagenic substances, including heavy metals and radiation [80,81,82]. Fish specimens in the current study were collected from three geographic and ecologically distant locations. Lake Brullus is a relatively shallow and confined northern brackish water lake. On the other hand, EL-Qanater is considered a middle location belonging to the river Nile, while Lake Nasser is a massive freshwater southern man-made reservoir [65,66,67,68,69,83,84,85,86,87].

The assessment of SNP frequency in the current study revealed that Lake Nasser and EL-Qanater populations (MA, FA, MQ, and FQ) harbored the highest number of SNPs affecting body weight and length. This pattern was also reflected in the assessment of body measurements, as they showed the highest body characteristics compared to Lake Brullus populations.

The elevated SNP frequency of the EL-Qanater population is expected; hence, this population exists within the River Nile system with uncontrolled population migration. The cross-breeding among different populations enhances genetic diversity and subsequently increases the frequency of SNPs associated with improved body weight and length [80,88].

The high SNP frequencies were also reported in the Lake Nasser *O. niloticus* population in the current study, which could be attributed to Lake Nasser’s enormous surface area and significant depth variation across the lake (varying from five meters near the edge to 130 m near the dam). Also, the existence of 85 khors (flooded valleys) contributed to separating microenvironment formation with diverse heat and oxygen stratification profiles, which in turn trigger mutations, including SNPs for better adaptation [89]. On the other hand, the notably low genotype frequency of SNPs in fish recovered from Lake Brullus may be linked to inbreeding; hence, the lake’s shallow conditions promote inbreeding and decrease the occurrence of heterozygous individuals [83,84]. Furthermore, the limited gene flow due to physical barriers could potentially lead to a lower frequency of SNPs in growth hormone genes, where the diminished depth creates a relatively homogenous environmental condition across the lake [83,84,90,91]. Moreover, the massive inflow of pollutants into the lake through agricultural and industrial drainage channels causes detrimental impacts on water quality and fish health [80,91,92]. These preceding factors result in oxidative stress, DNA damage, and heightened selection pressures linked to tolerance and survival rather than improved growth performance [92]. The presence of an increased number of genotypes with SNPs within a specific population(s) would suggest their suitability for body traits improvement when included in mating design for prospective development plans [93,94]. Therefore, for the anticipated hybridization step of the program, only Lake Nasser (A) and EL-Qanater (Q) populations are considered.

In the current study, the analysis of the *GH* gene (≈600 bp) identified 14 SNPs with a minor allele frequency (MAF) >1%. In addition, based on STRUCTURE analysis, population admixture has been detected, where the studied populations (MA, MB, MQ, FA, FB, and FQ) belonged to two presumed populations (*K* = 2) with no distinct differentiation between populations. These findings could imply the presence of a shared ancestor and/or similar selective force across populations [95]. Similarly, significant population admixture was previously reported via SNP-based structure analysis in *O. niloticus* strains cultured in Tanzania [96], where the examined populations originated from a single ancestor located at Lake Victoria, and this admixture was attributed to the uncontrolled movement of fish between different locations. A similar case was also reported by Fagbémi et al. [97] where only four populations, rather than seven, were present among seven wild and domesticated populations of *O. niloticus* from Benin, West Africa.

The present association analysis results highlight the complex genetic structure underlying growth performance in the studied populations. These traits were significantly associated with nine SNPs and three InDels, with *R*^2^ ranging from 2.6 to 36%. The wide range of phenotypic variance (*R*^2^) suggests the presence of a higher impact of some genetic variants on growth performance.

The study also revealed a correlation between only two SNPs (SNP1 and SNP8) and two InDels (InDel2 and InDel3) with both body weight and length, indicating the valuable application of such specific genetic markers in the next genetic improvement programs.

Haplotype blocks are segments of the chromosome characterized by elevated linkage disequilibrium (LD), diminished haplotype diversity, and a low rate of recombination [57,58]. Therefore, tightly linked SNPs (SNP1 and SNP2 block 1; SNP3, SNP4, SNP5, and SNP6 block 2; SNP7, SNP8, and SNP9 block 3) within a single haplotype block ensure their stability through generations (as they are inherited as cohesive units through generations). Such a pattern highlights SNPs’ importance as reliable markers for MAS in *O. niloticus* genetic improvement programs [98]. Similarly, Jaser et al. [73] reported comparable findings where a total of 10 SNPs located within the promoter and first intron of the *GH* gene in *O. niloticus* showed significant association with increased body weight. They attributed this pattern to the genetic background of the examined fish, in addition to the different linkage disequilibrium (LD) extent, which in turn influenced the formation of haplotype blocks.

The identified SNPs, during the current study, were significantly associated with a high growth rate, allowing the future use of such SNPs in MAS programs. In the same context, Cuevas-Rodríguez et al. [99] found two SNPs in the promoter area of the *GH* gene, eight SNPs in the introns and promoter region of the *IGF-I* gene, and one SNP in the myogenin gene linked to growth in Nile tilapia, *Oreochromis niloticus*. Similarly, Li et al. [100] assessed the marker-trait associations of growth in yellow catfish, *Pelteobagrus fulvidraco*, through direct sequencing of the growth hormone gene. They discovered three SNPs significantly associated with growth traits. Moreover, Naby et al. [101] reported the presence of growth-related SNPs within the intron and exon of the *GH* receptor gene in Nile tilapia, *Oreochromis niloticus*.

The variation in SNP frequencies, in the current study, was also noticed to be sex-related; the male population showed higher SNP frequencies compared to females, where MQ had six SNPs and one InDel, while MA showed five SNPs and one InDel. In contrast, females (FA and FQ) had two SNPs and one InDel, which were associated with both length and weight. This may be explained by the effect of natural selective pressure, where larger males with faster growth rates due to *GH* polymorphisms (SNPs) have higher chances for survival and the advantage in male-to-male competition during breeding seasons, resulting in a higher chance of mating with females [78,102].

The larval stage represents the most critical stage of the aquaculture cycle, as the growth performance in early life stages of fish affects their overall growth performance [103]. Furthermore, at this stage, the immune system of larvae is undeveloped; therefore, they are highly vulnerable to environmental stressors and infectious agents [26,104,105].

The gene expression assessment can directly detect changes resulting from different genomic signatures, including variations in allele frequencies, promoter modification, and heritable epigenetic alteration, even after a single generation of selection [106].

In the current study, the expression profiles of 72-hour-old larvae were studied utilising comparative threshold cycle (CT). The results were recorded as fold change and compared to the control after normalisation by the *β-actin* gene as a reference gene. While the use of multiple reference genes enhances the accuracy and stability of qPCR normalisation, several studies have validated the suitability of *β-actin* as a reliable single reference gene in *Oreochromis niloticus* under various experimental conditions [107,108]. Therefore, the use of a single, well-validated housekeeping gene such as *β-actin* remains acceptable and scientifically justified in gene expression studies, particularly where resource limitations exist [69,75,104].

In the current study, the *IGFI* was significantly upregulated in all the tested progeny compared to the control group, while the *GH* gene expression was at its highest in Lake Nasser progeny compared to all remaining groups; considering that all larvae were reared under identical environmental conditions, their elevated growth-related genes might be referred to their genetic background. The observed synergistic upregulation of *GH* and *IGF1* gene expression in Lake Nasser (A) progeny is likely due to the regulatory influence of the *GH* gene on *IGF1*. *GH* gene upregulation triggers several signaling pathways involving Janus kinase 2 (*JAK2*), signal transducers and activators of transcription (STAT), phosphatidylinositol 3-kinase (*PI3K*)-protein kinase B (*AKT*), and extracellular signal-regulated kinase (ERK). These pathways ultimately lead to enhanced transcription of *IGFI*, explaining its increased expression in response to *GH* stimulation [26,109]. Similarly, Opazo et al. [110] studied the variation in gene expression of 20-day-old larvae divided according to their growth performance into two groups (30 specimens each), large and small; these larvae were obtained from one pair of adult zebrafish and reared under the same environmental conditions. Their finding revealed a significant difference in the gene expression of *GH* and *IGFI* between the fast- and slow-growing siblings, indicating a genetic basis for growth variation. Additionally, Devlin et al. [111] have reported the role of genetic elements in growth performance alteration, where they reported differences in gene expression of *GH* and *IGFI* genes in three juvenile populations (wild, domesticated, and transgenic) of roho salmon (*Oncorhynchus kisutch*) maintained at identical conditions. Furthermore, Christie et al. [106] examined the gene expression profile of multiple genes in offspring of wild and hatchery steelhead trout (*Oncorhynchus mykiss*) reared in a common environment; their results reported a significant gene expression alteration in 723 genes, including growth-related genes. They referred their finding to the massive, heritable changes occurring during the first stages of selection and domestication.

In teleost, *GH* shows an evident positive connection to immunity, as the elevation of its circulating level improves immunity and increases leukocyte proliferation, while hypophysectomy is linked to immune suppression [112,113,114].

Phagocytosis (innate immune response) activation and regulation are driven by proinflammatory cytokines, especially *IL1β* and *TNFα*, in addition to chemokines (*CXC_2_* and *CC*), small chemoattractant proteins. The synergetic action of cytokines and chemokines creates a robust immune network, enabling swift and localised immune responses [115].

In the current study, the gene expression of *IL1β* and *IL8* was significantly upregulated in Lake Nasser (A) and EL-Qanater (Q) progenies compared to Lake Brullus (B) and the control groups. The *CXC_2_* RNA transcript was significantly upregulated in the progeny of the three populations compared to the reference group, while the *CC* chemokine and *TNFα* show no significant difference compared to the control. Likewise, Pontigo and Vargas-Chacoff [116] recorded the significant upregulation of *IL1β* and *IL8* gene expression 16 h post-stimulation with growth hormone in two Atlantic salmon (*Salmo salar*) head kidney cell lines. Furthermore, Shved et al. [117] reported a positive relationship between elevated growth hormone and upregulation of *TNFα*, where they reported a significant induction of RNA transcript of *TNFα* in the spleen and pituitary gland of Nile tilapia upon intraperitoneal injection with growth hormone. Moreover, Nakharuthai and Srisapoome [118] documented the induction of *CXC_2_* gene expression in unstressed Nile tilapia; they referred to this induction as the immune surveillance activity of *CXC_2_* chemokine.

The gene expression of the *sacs* gene in the current study was significantly upregulated in all tested larval groups, while the gene expression of *Rag* was significantly upregulated in only Lake Nasser and EL-Qanater offspring. This result could imply an early development of both the immune and neural immune systems, as the *Rag* gene is related to the early development of the adaptive immune response (B and T cells) of teleost [62], where it encodes proteins related to V(D)J recombination, a process necessary for diverse antigen combinations by lymphocytes [119,120]. In contrast, the *sacs* gene is related to early neural development and mitochondrial integrity [120]. Similarly, Lee et al. [121] tracked the gene expression pattern of the *Rag* gene starting from five to fifty-five days post-hatch for tracking the developmental stages of the adaptive immune system in olive flounder, *Paralichthys olivaceus*. Furthermore, Lee et al. [122] examined the gradual development of the immune system through *Rag* and *sacs* genes expression assessment in loach (*Misgurnus anguillicaudatus*) from day zero post-hatching to 40 days post-hatching.

## 5. Conclusions

The present study on Nile tilapia, *Oreochromis niloticus*, a key species in global and Egyptian aquaculture, provides valuable insights into the genetic associations of body traits with single-nucleotide polymorphisms (SNPs) in the growth hormone (*GH*) gene. Furthermore, the effect of different ecological conditions and sex on SNP abundance was recorded. The population structure revealed the presence of two homogeneous groups, with no clear separation between the studied populations. Association analysis revealed the presence of nine SNPs and three InDels with significant effects on body weight and length. Some of the identified SNPs are simultaneously associated with an increase in body weight and body length, recognized as the most valuable SNPs (SNP1, SNP8, InDel2, and InDel3). Additionally, population-specific SNPs were also detected among the studied populations, suggesting that the use of such populations would accelerate tilapia breeding programs. In all locations, SNP frequencies differed between male and female populations, highlighting the variance in natural selection pressure in opposite sexes under the same circumstances due to behavioral and hormonal differences. Linkage disequilibrium analysis identified three main haplotype blocks, facilitating further genomic analyses. The gene expression assessment of growth, proinflammatory, immunological, and neurological development-related genes in F1 larvae of the selected parent revealed a significant improvement of all tested parameters. Notably, the significant induction of the RNA transcripts of the growth-related genes *GH* in the F1 progeny of Lake Nasser and *IGFI* across all selected groups (B, A, and Q) might indicate that the identified SNPs are not only statistically associated with growth traits but also function biologically. Furthermore, the simultaneous induction of proinflammatory cytokines, chemokines, and genes related to early immune and neural development in the F1 progeny suggests a harmonised molecular response that may enhance early survival, physiological resilience, and overall growth potential. The current research highlights the potential for utilizing specific populations harboring candidate SNPs in prospective tilapia breeding programs to enhance desirable traits.

## Figures and Tables

**Figure 1 animals-15-02097-f001:**
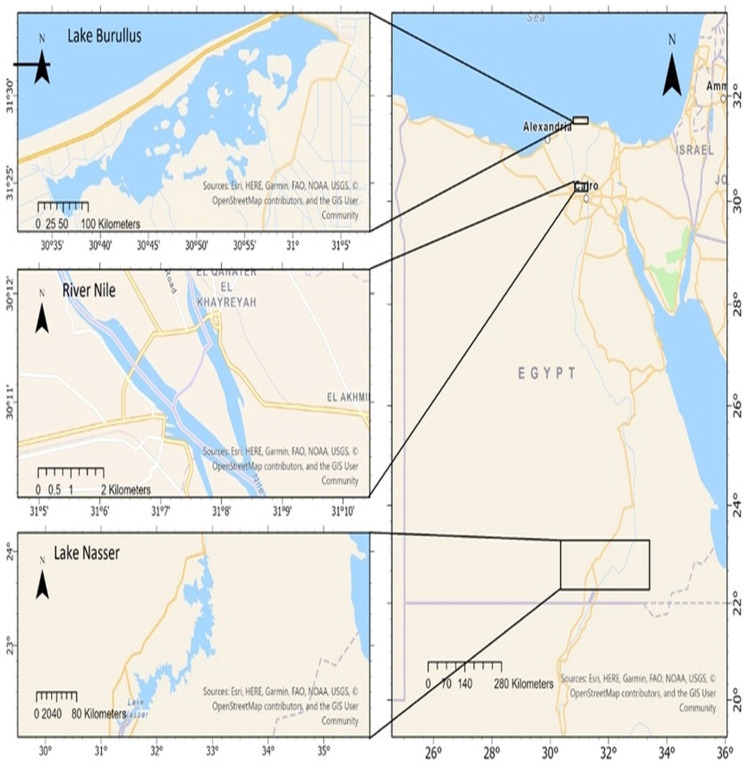
Nile Tilapia sampling locations: Lake Brullus, El-Qanater (River Nile), and Lake Nasser, taken after [32].

**Figure 2 animals-15-02097-f002:**
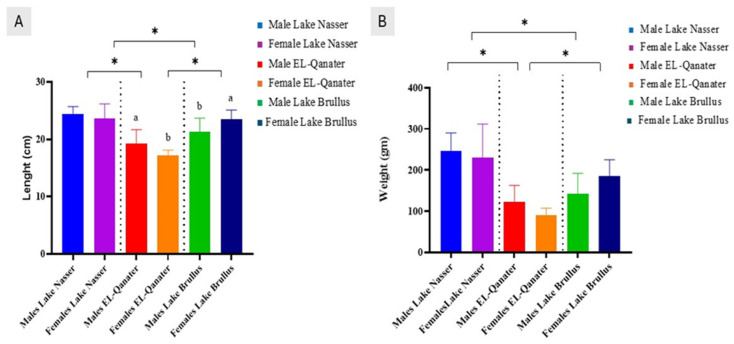
Comparison of body length (**A**) and body weight (**B**) among male and female Nile tilapia (*Oreochromis niloticus*) from three different populations: Lake Nasser, El-Qanater, and Lake Brullus. Bars represent the mean ± SE. (*) indicate significant differences between groups based on a two-way ANOVA followed by Tukey’s post hoc test (*p* ≤ 0.05). Different lowercase letters above bars in panel A denote significant differences within the opposite sexes across locations.

**Figure 3 animals-15-02097-f003:**
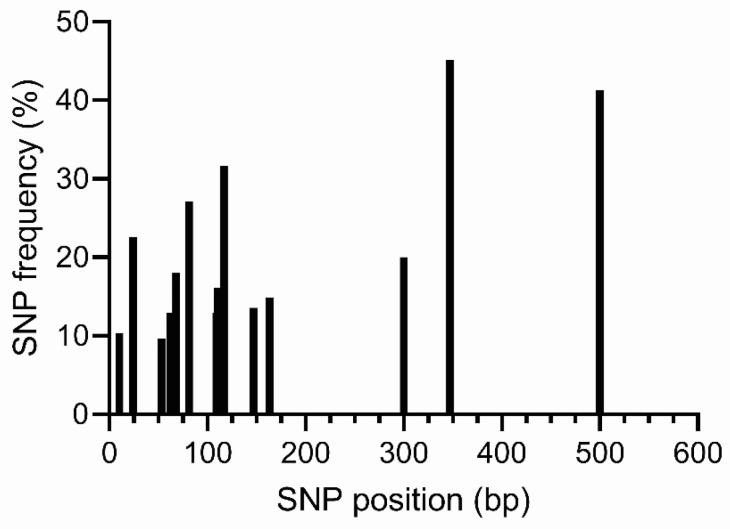
SNPs with MAF > 1% within *GH* gene across 155 genotypes from six populations of Nile Tilapia.

**Figure 4 animals-15-02097-f004:**
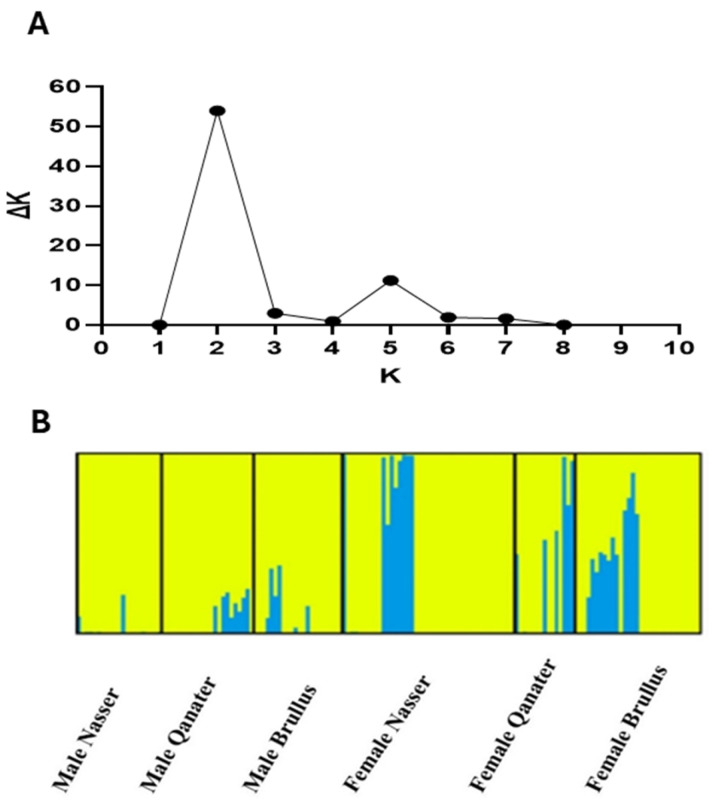
SNP-based STRUCTURE analyses of Nile tilapia male and female genotypes: (**A**) The magnitude of ∆*K* as a function of *K* (i.e., the number of defined groups). (**B**) SNP-based STRUCTURE defined groups among six populations for *K* = 2. Yellow and blue are colors representing group 1 and group 2.

**Figure 5 animals-15-02097-f005:**
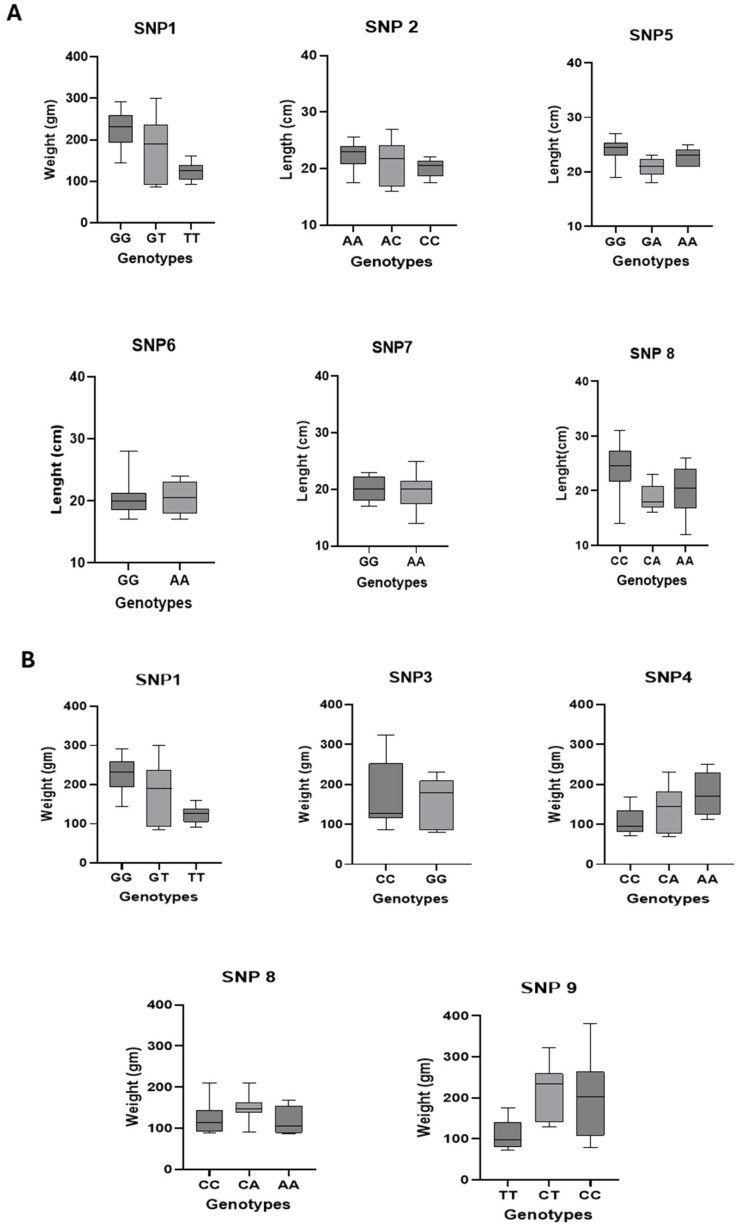
Boxplot diagrams depicting the genetic effects of SNPs with significant association with (**A**) body length and (**B**) body weight in Nile tilapia.

**Figure 6 animals-15-02097-f006:**
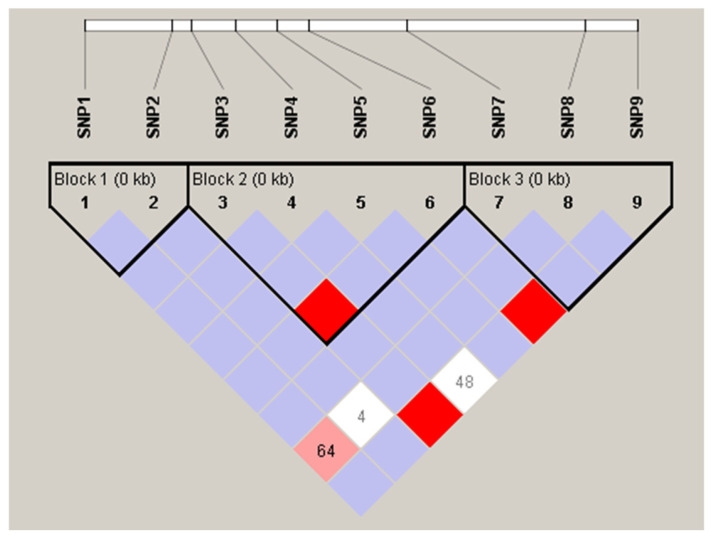
Haplotype block map for all nine SNPs. Linkage disequilibrium (LD) plots containing nine SNPs from the *GH* gene of Nile tilapia. LD is measured as D’, ranging from 0 to 1. D’ value equals 1 is depicted in red, and less than 1 is depicted in shades of pink/light red, light blue, and white. The black lines mark the identified blocks.

**Figure 7 animals-15-02097-f007:**
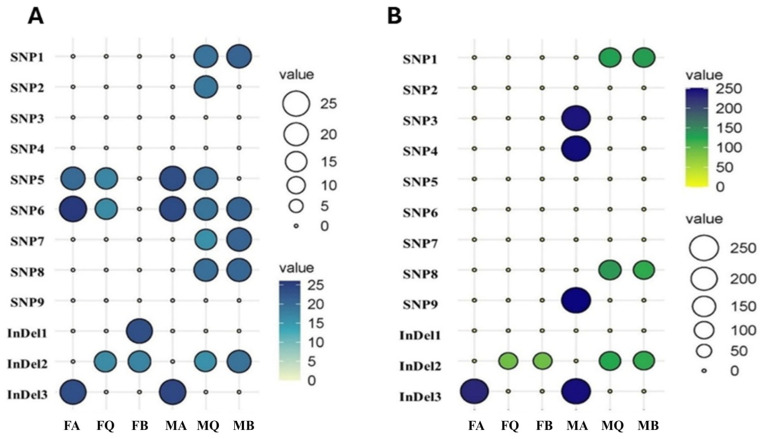
Ballon plot of the effect of SNP/InDel on the average body length (**A**) and body weight (**B**) in six broad stock populations of Nile tilapia. Females Lake Nasser (FA), females EL-Qanater (FQ), females Lake Brullus (FB), males Lake Nasser (MA), males EL-Qanater (MQ), and males Lake Brullus (MB).

**Figure 8 animals-15-02097-f008:**
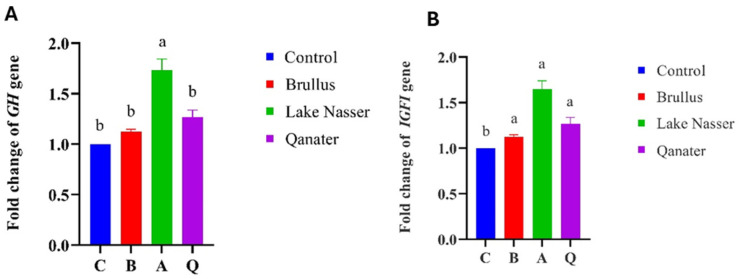
Gene expression of growth-related genes, growth hormone (*GH*) (**A**), and insulin growth factor I (*IGFI*) (**B**) of *Oreochromis niloticus* larvae 72 h post-hatching. C is the control, B is Lake Brullus, A is Lake Nasser, and Q is El-Qanater F1. Different lowercase letters above bars denote significant differences between different larval groups (*p* < 0.05).

**Figure 9 animals-15-02097-f009:**
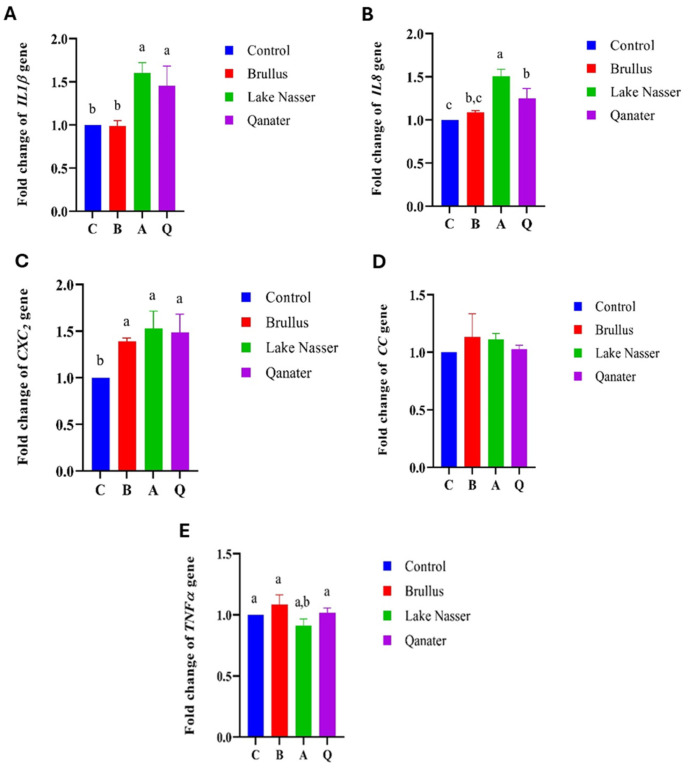
Gene expression of proinflammatory-related genes, interleukin 1 beta *(IL1β)* (**A**), interleukin 8 (*IL8*) (**B**), CXC2-chemokine (*CXC2*) (**C**)*,* CC-chemokine (*CC*) (**D**), and tumor necrosis factor alpha (*TNFα*) (**E**) in *Oreochromis niloticus* larvae 72 h post-hatching. C is the control, B is Lake Brullus, A is Lake Nasser, and Q is El-Qanater F_1_. Different lowercase letters above bars denote significant differences between different larval groups (*p* < 0.05).

**Figure 10 animals-15-02097-f010:**
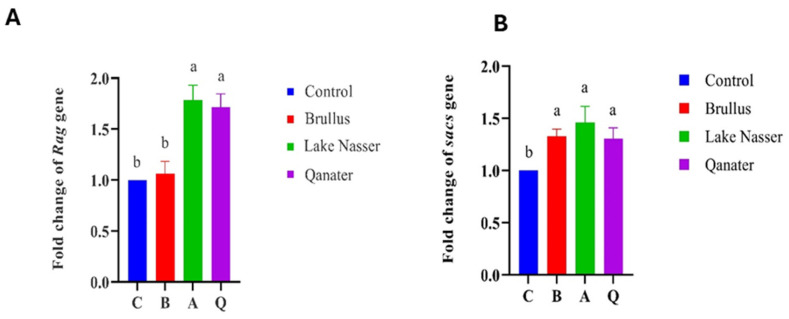
Gene expression of early immune system development-related gene, recombinant reactivating gene *(Rag*) (**A**), and early neural development-related gene, sacsin protein (*sacs*) (**B**) in *Oreochromis niloticus* larvae 72 h post-hatching. C is the control, B is Lake Brullus, A is Lake Nasser, and Q is El-Qanater F_1_. Different lowercase letters above bars denote significant differences between different larval groups (*p* < 0.05).

**Table 1 animals-15-02097-t001:** Code, forward, and reverse sequences of primers utilized in quantitative real-time PCR.

Gene Name and Code	Forward Primer (5′-3′)	Reverse Primer (5′-3′)	Target Size (bp)	Linkage Group	Accession Number	Reference
*CXC*_2_ chemokine	CTATCCATGGAGCCTCAGGT	CTTCTTGAGCGTGGCAATAA	146	LG6	XM_003452201.5	[59]
Interleukin-1 Beta (*IL1β*)	AGAGCAGCAATTCAGAGC	GTGCTGATGTACCAGT	514	LG12	XM_019365842.2	[60]
Interleukin-8 (*IL8*)	GCACTGCCGCTGCATTAAG	GCAGTGGGAGTTGGGAAGAA	85	LG6	XM_019359413.2	[61]
CC-chemokine	ACAGAGCCGATCTTGGGTTACTTG	TGAAGGAGAGGCGGTGGATGTTAT	228	LG9	XM_019362411.2	[60]
“Recombination Activating Gene1 (*rag1*)”	AAGACAGTGCCTGCACATCA	CTCAGGAACAACTGGTCCCC	104	LG7	XM_019361952.2	[62]
Sacsin gene (*sacs*)	ACATCACTGATAGCACCCGC	TCATGGTGGGATCTGGACCT	82	LG14	XM_019367592.2	[62]
Tumer necrosis factorα (*TNFα*)	GGTTAGTTGAGAAGAAATCACCTGCA	GTCGTCGCTATTCCCGCAGATCA	407	LG11	NM_001279533.1	[63]
Insulin like growth factor1 (*IGF1*)	TTGTCTGTGGAGAG CGAGGCT	CAGCTTTGGAAGCA GCACTCGT	202	LG17	XM_019346352.2	[60]
Growth hormone (*GH*)	TAATGGGAGAGGGA AGATGG	CTCTGCGATGTAAT TCAGGA	90	LG4	XM_003442542	[64]
*βeta-actin*	TGG CAA TGA GAG GTT CCG	TGCTGTTGTAGGTGGTTTCG	136	LG4	XM_003443127.5	[65]

**Table 2 animals-15-02097-t002:** Associations between SNPs/indels and body length and weight in *GH* gene of Nile tilapia showing the SNP number, and its attributable % of phenotypic variance (*R*^2^). Listed associations were significant at *p* ≤ 0.05 and were verified by the false discovery rate method.

Attribute	SNP/InDel	Position in Consensus Sequence	*p*-Value	% Phenotypic Variance (*R*^2^)
Body length	SNP1 *	36	0.006	4.9
	SNP2	116	0.02	3.5
	SNP5	211	0.016	3.8
	SNP6	240	0.015	3.8
	SNP7	330	0.008	4.5
	SNP8 *	492	0.007	4.7
	InDel1	572	0.005	4.9
	InDel2 *	581	0	14.8
	InDel3 *	582	0	23.4
Body weight	SNP1 *	36	0.028	3.1
	SNP3	134	0.042	2.6
	SNP4	174	0.045	2.6
	SNP8 *	492	0.029	3.1
	SNP9	539	0.028	3.1
	InDel2 *	581	0	18.2
	InDel3 *	582	0	36.0

* Common associations with body length and body weight.

**Table 3 animals-15-02097-t003:** Number of genotypes per population with significant SNPs and InDels identified in the *GH* gene in six *Nile tilapia* populations.

SNP/InDel	FA	FQ	FB	MA	MQ	MB
SNP1					8	1
SNP2					4	
SNP3				6		
SNP4				5		
SNP5	1	1		9	5	
SNP6	1	2		7	5	6
SNP7					2	3
SNP8					6	1
SNP9				6		
InDel1			8			1
InDel2		8	1		8	1
InDel3	10			6		
Total	12	11	9	39	38	12

Females Lake Nasser (FA), females EL-Qanater (FQ), females Lake Brullus (FB), males Lake Nasser (MA), males EL-Qanater (MQ), males Lake Brullus (MB).

## Data Availability

The datasets used and/or examined during the current research are available from the corresponding author upon reasonable request.

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
