# Peer review of "Selection for Growth Performance in Oreochromis niloticus Across Different Aquatic Environments Using Growth Hormone Gene Polymorphisms"

_animals, 2025, doi:10.3390/ani15142097_

Round 1
Reviewer 1 Report
Comments and Suggestions for Authors
The main concern involves the so-called carryover effects (i.e., the pre-experiment environment) on the phenotypic data of the three different populations (stocks). These effects may have introduced bias in the GWAS results or led to false-positive associations between SNPs and growth traits.
Specific comments focusing on the Materials and Methods are as follows:
Heading 2.2:
Specify the sex distribution by stock (i.e., number of females and males per stock).
Provide details on the experimental or culture period in the tanks. Include tank sizes, tank conditions, and the exact number of fish (females and males per stock per tank).
Heading 2.3:
Clarify the purpose of fry collection and describe the fry rearing procedures.
Heading 2.4:
Provide the exact number of individuals per stock for which DNA samples were collected and sequenced.
Specify the criteria used for quality control (QC) of raw sequence data.
Detail the parameter settings used for variant (SNP) calling in the software.
Describe the QC procedures for SNPs, including the exact number of SNPs before and after QC.
Outline the characteristics or properties of the final SNP set used for downstream analyses.
Include the accession number for the partial GH sequences used to generate a consensus sequence.
Indicate the optimal k value used.
Clearly describe the specific fixed and random effects included in the mixed model.
Clarify the multiple testing procedures used via FDR, including how FDR was calculated. Also, specify the exact –log₁₀(P-value) significance threshold.
Specify the window sizes of chromosomal regions used in the LD analysis.
Heading 2.5:
What were the control samples?
An additional reference gene is needed to ensure the normality and accuracy of the qPCR data.
Add to Table 1 the chromosomal locations or genomic regions of all the genes used for qPCR analysis.
Elaborate on the acquisition and normalization of the qPCR data.
Were any biological replicates used in the qPCR reactions?
Heading 2.6:
Statistical analyses should be described specifically for each experiment, data type, or variable. Most importantly, the association analysis must account for both fixed (e.g., population, location, age, initial weight, and sex of individuals) and random effects on the studied phenotypes.
Additional suggestion: Consider performing a combined analysis using both SNP and gene expression data.
Author Response
Response to Reviewer 1 Comments
|
|||
1. Summary |
|
|
|
We would like to express our sincere appreciation to the reviewers for their thoughtful evaluation of our manuscript titled “Selection for growth performance in Oreochromis niloticus across different aquatic environments using growth hormone gene polymorphisms.” We are truly grateful for the reviewer’s detailed observations and constructive suggestions, which have helped us improve the clarity, depth, and overall quality of the manuscript. We have carefully addressed all comments, and the corresponding revisions have been clearly highlighted in the resubmitted manuscript. Please find our detailed point-by-point responses below.
|
|||
2. Questions for General Evaluation |
Reviewer’s Evaluation |
Response and Revisions |
|
Does the introduction provide sufficient background and include all relevant references? |
Must be improved |
We added some modifications |
|
Are all the cited references relevant to the research? |
Must be improved |
We added some modifications |
|
Is the research design appropriate? |
Must be improved |
Improved
|
|
Are the methods adequately described? |
Must be improved |
Improved
|
|
Are the results clearly presented? |
Must be improved |
Improved |
|
Are the conclusions supported by the results? Are all figures and tables clear and well-presented? |
Must be improved
Must be improved
|
We added some modifications |
|
3. Point-by-point response to Comments and Suggestions for Authors |
|||
Comments 1: The main concern involves the so-called carryover effects (i.e., the pre-experiment environment) on the phenotypic data of the three different populations (stocks). These effects may have introduced bias in the GWAS results or led to false-positive associations between SNPs and growth traits.
|
|||
Response 1: We agree that in studies involving wild-caught fish, it is crucial to address any pre-experimental differences in environmental exposure that could confound marker-trait relationships. To minimize such effects, we implemented the following strategies: 1- All 155 fish representing the three wild Nile tilapia populations were subjected to a 30-day acclimatization period under controlled and identical rearing conditions at the National Institute of Oceanography and Fisheries (El-Qanater branch). These measures aimed to neutralize the influence of prior environmental exposures and synchronize metabolic baselines [1–3]. 2- Growth, body weight and length were measured monthly over the experiment period following acclimatization. This allowed us to capture dynamic growth patterns under unified environmental conditions to ensure that the phenotypic data used for association analysis reflects inherent genetic potential, rather than residual environmental conditioning. 3- Marker-Trait Association under uniform conditions: to mitigate such confounding influences from carryover effects, we implemented a mixed linear model (MLM) approach for association mapping, which explicitly accounts for population structure (Q matrix) and kinship (K matrix). This model allowed for accurate estimation of SNP effects while controlling for confounding effects due to relatedness and environmental variables. Adjusting for these covariates, the model increases the robustness of SNP-trait associations and minimizes the likelihood that observed signals are due to environmental or population-specific artifacts rather than true genetic effects. 4- The association mapping was conducted using these growth traits recorded post-acclimation and during uniform rearing, to identify genetic variants (SNPs and InDels) within the GH gene correlated with body size. This controlled environment minimizes genotype-by-environment interaction and ensures that SNP-trait associations are biologically meaningful.
Comparable published designs: Our methodology is further supported by similar designs in other wild fish association studies, where phenotypes are measured in the wild or shortly after capture, for example,[1–3]
|
|||
Comments 2: Specify the sex distribution by stock (i.e., number of females and males per stock). |
|||
Response 2: Done, a complete clarification of the number of females and males obtained from each location has been included (lines 124-125 and 130-134). |
|||
Comments 3: Heading 2.2: Specify the sex distribution by stock (i.e., number of females and males per stock). Provide details on the experimental or culture period in the tanks. Include tank sizes, tank conditions, and the exact number of fish (females and males per stock per tank).
Response 3: Done, the sex distribution by stock (i.e., number of females and males per stock) and detailed cultural conditions have been incorporated in heading 2.2 of the material and methods section (lines 124-125, 130-134, 140-143, and 149-151).
|
|||
Comments 4: Heading 2.3: Clarify the purpose of fry collection and describe the fry rearing procedures. Response 4: Done, a section has been incorporated into the material and methods heading 2.3 with necessary references (lines 152–161).
|
|||
Comments 5: Heading 2.4: Provide the exact number of individuals per stock for which DNA samples were collected and sequenced. Response 5: Done, the exact number of individuals per stock for which DNA samples were collected and sequenced has been added (lines 168-170).
Comment 6: Specify the criteria used for quality control (QC) of raw sequence data. Response 6: Done, a section was incorporated in heading 2.4.2 (lines 187-194) to clarify the criteria used for quality control (QC) of raw sequence data.
Comment 7: - Detail the parameter settings used for variant (SNP) calling in the software. Response 7: Done, details of the parameter settings used for variant (SNP) calling has been included in heading 2.4.2 (lines 201-205).
Comment 8: Describe the QC procedures for SNPs, including the exact number of SNPs before and after QC. Response 8: Done, a statement of QC procedures was included in heading 2.4.2 (lines 188-195) and (lines 201-205). The exact number of SNPs before and after QC procedures is updated in the results part, heading 3.2 (line 314-322).
Comment 9: Outline the characteristics or properties of the final SNP set used for downstream analyses. Response 9: Done, a statement of the characteristics of the final SNPs set used for downstream analyses is included in the results part, heading 3.2 (lines 321-323).
Comment 10: Include the accession number for the partial GH sequences used to generate a consensus sequence. Response 10: The accession numbers for the partial GH gene sequences used to generate the consensus sequence have been provided in the manuscript (accession number LC832463 to LC832617, heading 2.4.2 (line 206-207), representing the full set of submitted sequences. We have ensured their visibility and clarity in the revised version.
Comment 11: Indicate the optimal k value used. Response 11: The optimal K used is K=2, heading 3.3 (line 332-333).
Comment 12: Clarify the multiple testing procedures used via FDR, including how FDR was calculated. Also, specify the exact –log₁₀(P-value) significance threshold. Response 12: Done, a detailed statement of multiple testing procedure used via FDR based on Benjamini-Hochberg with a specific P-value is included in heading 2.4.4 (lines 237-243).
Comment 13: Specify the window sizes of chromosomal regions used in the LD analysis. Response 13: Done, the window size of 300 bp used in the LD analysis is included in heading 2.4.5 (lines 246-247).
Comment 14: Heading 2.5: What were the control samples? Response 14: Done, clarification of the control group has been included lines (252-255).
Comment 15: An additional reference gene is needed to ensure the normality and accuracy of the qPCR data. Response 15: We appreciate the reviewer’s concern regarding the stability and normalization of qPCR data. However, in the present study, we used β-actin as the internal reference gene for relative expression analysis, which is widely accepted and validated for gene expression studies in Nile Tilapia and other teleost fish. Numerous recent peer-reviewed studies in Nile tilapia have demonstrated the stability and sufficiency of β-actin as a single reference gene, including [4–7].
Comment 16: Add to Table 1 the chromosomal locations or genomic regions of all the genes used for qPCR analysis. Response 16: Done, a column of linkage group of each gene utilized in real time PCR has been incorporated in Table 1
Comment 17: Elaborate on the acquisition and normalization of the qPCR data. Response 17: Done, a section has been added to heading 2.6 (lines 286- 297).
Comment 18: Were any biological replicates used in the qPCR reactions? Response 18: Yes, three larvae per group were used as replicate. This is clarified in heading 2.5.1 (line 253), and the reaction was repeated twice. This is clarified in heading 2.5.2 (line 275).
Comment 19: Statistical analyses should be described specifically for each experiment, data type, or variable. Most importantly, the association analysis must account for both fixed (e.g., population, location, age, initial weight, and sex of individuals) and random effects on the studied phenotypes. Response 19: Done, a clarification of association analysis model is incorporated in heading 2.4.4 (lines 225-233). Clarification of statistical analyses of other experiments has been also incorporated in heading 2.6 (lines 283-297). |
|||
Comment 20: Additional suggestion: Consider performing a combined analysis using both SNP and gene expression data. Response 20: We are grateful for the reviewer’s suggestion and understand the scientific value of integrating both genotypic (SNP) and transcriptomic (gene expression) data, as it might provide a more comprehensive understanding of the functional consequences of GH gene polymorphisms and their regulatory impact on growth and immune traits in Nile tilapia. However, due to the current scope and structure of our dataset, the integration was beyond the objectives of our manuscript. However, we recognize its scientific merit and plan to incorporate this approach in our future work.
|
|||
Response to Reviewer 2 Comments
|
||||||
1. Summary |
|
|
||||
We sincerely thank you for your time, effort, and constructive feedback on our manuscript titled “Selection for growth performance in Oreochromis niloticus across different aquatic environments using growth hormone gene polymorphisms.” We greatly appreciate the valuable insights and thoughtful suggestions, which have helped us improve the clarity, scientific rigor, and overall quality of the manuscript. Please find below our detailed responses to each comment, along with the corresponding revisions clearly highlighted in the resubmitted version of the manuscript.
|
||||||
2. Questions for General Evaluation |
Reviewer’s Evaluation |
Response and Revisions |
||||
Does the introduction provide sufficient background and include all relevant references? |
yes |
|
||||
Is the research design appropriate? |
can be improved |
Improved |
||||
Are the methods adequately described? |
yes |
|
||||
Are the results clearly presented? |
yes |
|
||||
Are the conclusions supported by the results? Are all figures and tables clear and well-presented? |
yes yes
|
|
||||
3. Point-by-point response to Comments and Suggestions for Authors |
||||||
|
|
|
|
|||
|
Comments 1: The number and sex ratio of each population are not mentioned in the manuscript. This is important as there are significant size differences between male and female Oreochromis niloticus of the same age. Response 1: Done, the exact number and sex ratios of tested population have been specified in heading 2.2 (lines 125-126 and 131-134). Comments 2: Although the three water bodies consist of one-year-old fish, their ages may still differ, which could contribute to the observed inter-group differences. Response 2: We agree that slight differences in hatch timing may exist, which could contribute to inter-population variability in growth traits. To address this, we applied several control measures: 1- First, fish were selected within a narrow range of body length and weight, corresponding to the one-year-old size class of Oreochromis niloticus. 2- Age verification was further supported by scale examination to ensure developmental consistency across samples [8]. 2- Longitudinal phenotyping by recording body weight and length monthly over the experimental period has been applied to minimize early growth variability and creates a more accurate representation of growth performance. 3- both location and sex were included as fixed effects in our statistical analysis, thereby reducing bias associated with developmental timing and population-specific differences.
Comments 3: The growth environments of the three wild populations are inherently different (as mentioned by the author in the discussion section), and the differences between the groups may be caused by environmental factors, with the SNPs potentially arising as adaptations to these environments Response 3: We agree that in studies involving wild-caught fish, it is crucial to address any pre-experimental differences in environmental exposure that could confound marker-trait relationships. To minimize such effects, we implemented the following strategies: 1- All 155 fish representing the three wild Nile tilapia populations were subjected to a 30-day acclimatization period under controlled and identical rearing conditions at the National Institute of Oceanography and Fisheries (El-Qanater branch). These measures aimed to neutralize the influence of prior environmental exposures and synchronize metabolic baselines. 2- Growth, body weight and length were measured monthly over the experiment period following acclimatization. This allowed us to capture dynamic growth patterns under unified environmental conditions to ensure that the phenotypic data used for association analysis reflects inherent genetic potential, rather than residual environmental conditioning. 3- Marker-Trait Association under uniform conditions: The association mapping was conducted using growth traits recorded post-acclimation and during uniform rearing, to identify genetic variants (SNPs and InDels) within the GH gene correlated with body size. This controlled environment minimizes genotype-by-environment interaction and ensures that SNP-trait associations are biologically meaningful. 4- Statistical control of population structure and relatedness: The association analysis was conducted using a mixed linear model (MLM) that incorporated both population structure and kinship relationship to account for any residual structure or family relatedness among individuals. This is a widely accepted approach to reduce false-positive associations. 5- Comparable published designs: Our methodology is further supported by similar designs in other wild fish association analysis, where phenotypes are measured in the wild or shortly after capture, for example, Johnston et al.[9] and Zhang et al. [10].
Comments 4: The significance of annotations in Figure 2.1 are unclear, especially for the inter-group differences in Figure 2.1A. It is recommended to add appropriate figure captions. Response 4: thank you for your constructive observation, the caption has been updated (lines 307-311). Comments 5: Why did the author choose the snp on the GH gene instead of other growth-related genes? Are there very few snp loci on these genes? Response 4: We thank the reviewer for this important question. The reason behind prioritizing SNPs within the growth hormone (GH) gene was detailed in the introduction section of the manuscript (lines 79-83). Briefly, GH plays a central regulatory role in the hypothalamic-pituitary-somatotropic (HPS) axis and is considered a master regulator of somatic growth in teleosts. It modulates a wide range of growth-related physiological processes, including muscle development, protein and energy metabolism, food conversion efficiency, sexual maturation, and immune function. Comments 5: The discussion section is too lengthy, making it difficult to pinpoint the key points, and it detracts from the focus of the paper. Response 5: We sincerely thank the reviewer for this observation. We acknowledge that the discussion is relatively detailed; however, this was intentionally structured to ensure that each subsection corresponds directly to the specific results presented, providing a clear interpretation for the biological relevance of each finding. Given the multidisciplinary nature of the study which integrates phenotypic, genetic, and gene expression data across three distinct populations, a comprehensive discussion was necessary to properly interpret the observed patterns and their implications in tilapia growth genetics. Each paragraph was tailored to clarify the relationship between individual results (e.g., SNP variation, expression profiles, and body metrics) and their biological significance within the broader framework of fish growth regulation. We ensure that the main findings are clearly summarized in the conclusion to enhance clarity and focus. That said, we remain open to condensing specific sections if it is necessary.
Comments 6: Finally, and most importantly, the purpose of this study is to explore growth-related SNPs for future breeding programs. Using three wild populations as research subjects seems unwise due to the many uncontrollable factors that contribute to their differences, despite the author's efforts to control these variables. It is suggested that using small-month-old F1 progeny from the three populations would be more appropriate. These F1 individuals inherit SNPs from their parents, and their growth can be controlled in a single environment, thereby eliminating environmental factors. This would make the contribution of SNPs to growth more convincing
Response 6: We appreciate the reviewer’s perspective, which strongly aligns with our ongoing research objectives. However, the present study was designed as an initial screening phase, aiming to identify SNPs in wild-derived broodstocks across genetically and ecologically distinct populations and to evaluate the early larval performance obtained from these selected broodstocks. Furthermore, for the next step of selective breeding of Nile tilapia, currently running at the National Institute of Oceanography and Fisheries, we aim to link these genetic signatures to their transgenerational effects by examining their inheritance and phenotypic impact in the F1 generation. We believe that this combined framework will offer a robust foundation for future selective breeding applications in Nile tilapia.
|
|||||
References:
- Zhang, C.; Zhang, Y.; Liu, C.; Wang, L.; Dong, Y.; Sun, D.; Wen, H.; Zhang, K.; Qi, X.; Li, Y. Genome‐wide Association Study and Genomic Prediction for Growth Traits in Spotted Sea Bass ( Lateolabrax Maculatus) Using Insertion and Deletion Markers. Animal Research and One Health 2024, 2, 400–416, doi:10.1002/aro2.87.
- Johnston, S.E.; Orell, P.; Pritchard, V.L.; Kent, M.P.; Lien, S.; Niemelä, E.; Erkinaro, J.; Primmer, C.R. Genome-Wide SNP Analysis Reveals a Genetic Basis for Sea-Age Variation in a Wild Population of Atlantic Salmon (Salmo Salar). Mol Ecol 2014, 23, 3452–3468, doi:10.1111/mec.12832.
- Zhang, C.; Wen, H.; Zhang, Y.; Zhang, K.; Qi, X.; Li, Y. First Genome-Wide Association Study and Genomic Prediction for Growth Traits in Spotted Sea Bass (Lateolabrax Maculatus) Using Whole-Genome Resequencing. Aquaculture 2023, 566, doi: 10.1016/j.aquaculture.2022.739194.
- Pholchamat, S.; Vialle, R.; Luang-In, V.; Phadee, P.; Wang, B.; Wang, T.; Secombes, C.J.; Wangkahart, E. Evaluation of the Efficacy of MONTANIDETM GR01, a New Adjuvant for Feed-Based Vaccines, on the Immune Response and Protection against Streptococcus Agalactiae in Oral Vaccinated Nile Tilapia (Oreochromis Niloticus) under Laboratory and on-Farm Conditions. Fish Shellfish Immunol 2024, 149, doi:10.1016/j.fsi.2024.109567.
- Ruenkoed, S.; Pholoeng, A.; Nontasan, S.; Panprommin, D.; Mongkolwit, K.; Wangkahart, E. Assessing the Impact of Acidifiers on Growth Performance, Innate Immune Capacity, Response to Ammonia Nitrogen Stress, Digestive Enzyme Activity, Intestinal Histology, and Gene Expression of Nile Tilapia (Oreochromis Niloticus). Fish Shellfish Immunol 2025, 162, doi: 10.1016/j.fsi.2025.110315.
- Tawfik, W.; Nassef, E.; Bakr, A.; Hegazi, E.; Ismail, T.A.; Abdelazim, A.M.; El-Nagar, S.H.; Sabike, I.; Fadl, S.E.; Sharoba, A.M. Orange Pulp in Nile Tilapia (Oreochromis Niloticus) Diets: Growth Performance, Biochemical Parameters and Gene Expression for Growth and Fat Metabolism. Aquac Rep 2022, 22, doi: 10.1016/j.aqrep.2021.100970.
- Awad, S.T.; Hemeda, S.A.; El Nahas, A.F.; Abbas, E.M.; Abdel-Razek, M.A.S.; Ismail, M.; Mamoon, A.; Ali, F.S. Gender-Specific Responses in Gene Expression of Nile Tilapia (Oreochromis Niloticus) to Heavy Metal Pollution in Different Aquatic Habitats. Sci Rep 2024, 14, 1–13, doi:10.1038/s41598-024-64300-4.
- Chebel, F.; Kara, H. Reproduction, Age and Growth of Tilapia Zillii (Cichlidae) in Oued Righ Wetland (Southeast Algeria); 2016; 30;40(3):235-43.
- Johnston, S.E.; Orell, P.; Pritchard, V.L.; Kent, M.P.; Lien, S.; Niemelä, E.; Erkinaro, J.; Primmer, C.R. Genome-Wide SNP Analysis Reveals a Genetic Basis for Sea-Age Variation in a Wild Population of Atlantic Salmon (Salmo Salar). Mol Ecol 2014, 23, 3452–3468, doi:10.1111/mec.12832.
- Zhang, C.; Zhang, Y.; Liu, C.; Wang, L.; Dong, Y.; Sun, D.; Wen, H.; Zhang, K.; Qi, X.; Li, Y. Genome‐wide Association Study and Genomic Prediction for Growth Traits in Spotted Sea Bass (Lateolabrax Maculatus) Using Insertion and Deletion Markers. Animal Research and One Health 2024, 2, 400–416, doi:10.1002/aro2.87.
Reviewer 2 Report
Comments and Suggestions for Authors
The author investigated SNPs in the growth hormone gene of Oreochromis niloticus based on three geographic populations, conducted a marker-trait association analysis, and associated the identified SNPs with growth performance (weight and length). Individuals with superior growth performance, carrying beneficial SNPs, were selected as the breeding foundation for future hybridization processes. The early performance of 72-day-old juveniles derived from the selected breeding population was also evaluated. This study is relatively comprehensive, with a large amount of work; however, several issues remain:
1.The author obtained 155 wild Oreochromis niloticus from three water bodies, with statistical analysis showing significant differences both between and within the groups. Before performing statistical analysis, some factors must be considered:
- ① The number and sex ratio of each population are not mentioned in the manuscript. This is important as there are significant size differences between male and female Oreochromis niloticus of the same age.
- ② Although the three water bodies consist of one-year-old fish, their ages may still differ, which could contribute to the observed inter-group differences.
- ③ The growth environments of the three wild populations are inherently different (as mentioned by the author in the discussion section), and the differences between the groups may be caused by environmental factors, with the SNPs potentially arising as adaptations to these environments.
2. The significance annotations in Figure 2.1 are unclear, especially for the inter-group differences in Figure 2.1A. It is recommended to add appropriate figure captions.
3. Why did the author choose the snp on the GH gene instead of other growth-related genes? Are there very few snp loci on these genes?
4. The discussion section is too lengthy, making it difficult to pinpoint the key points, and it detracts from the focus of the paper.
5. Finally, and most importantly, the purpose of this study is to explore growth-related SNPs for future breeding programs. Using three wild populations as research subjects seems unwise due to the many uncontrollable factors that contribute to their differences, despite the author's efforts to control these variables. It is suggested that using small-month-old F1 progeny from the three populations would be more appropriate. These F1 individuals inherit SNPs from their parents, and their growth can be controlled in a single environment, thereby eliminating environmental factors. This would make the contribution of SNPs to growth more convincing. Additionally, gene expression differences in growth and immune genes in small-month-old F1 progeny would likely be more pronounced and persuasive.
Author Response

(The authors gave the same response as above.)

Round 2
Reviewer 1 Report
Comments and Suggestions for Authors
Whatever measures were implemented, it is still not possible to account for the carryover effects on phenotypes. Therefore, this limitation should be discussed in the revised manuscript.
Second, include significance testing for the fixed effects: population, location, age, initial weight, sex, and SNP genotype, as a supplementary file.
Third, regarding qPCR: we were not questioning the use of β-actin as the internal reference gene. The question was whether an additional reference gene could be included to improve data normality and accuracy. If that is not possible, this limitation should also be discussed in the revised manuscript.
Finally, it is not good practice to cite studies with poor experimental design, especially as many of them have not undergone rigorous peer review.
Author Response
Response to Reviewer 1 Comments
|
||
1. Summary |
|
|
We are grateful to the reviewer for their ongoing assessment of our manuscript, "Selection for growth performance in Oreochromis niloticus across different aquatic environments using growth hormone gene polymorphisms." We hold the reviewer's revision and comments in high regard, as they have been instrumental in the enhancement of the quality and clarity of our work. We regret any misunderstanding or lack of clarity that may have been present in our previous responses. In this revision, we have taken great care to address all concerns more exhaustively. These clarifications are evident in the revised manuscript, which is plainly labelled with all modifications. We have provided a comprehensive, point-by-point response to each of the reviewer's comments below.
|
||
2. Questions for General Evaluation |
Reviewer’s Evaluation |
Response and Revisions |
Does the introduction provide sufficient background and include all relevant references? |
Can be improved |
Improved |
Are all the cited references relevant to the research? |
Can be improved |
Improved |
Is the research design appropriate? |
Can be improved |
Improved |
Are the methods adequately described? |
Can be improved |
Improved |
Are the results clearly presented? |
Can be improved |
Improved |
Are the conclusions supported by the results? |
Can be improved |
Improved |
3. Point-by-point response to Comments and Suggestions for Authors |
||
Comments 1: Whatever measures were implemented; it is still not possible to account for the carryover effects on phenotypes. Therefore, this limitation should be discussed in the revised manuscript. |
||
Response 1: Thank you for drawing attention to the limitations associated with carryover effects. In light of this, we have revised the manuscript to include a scientific discussion that reflects the current understanding: although our experimental protocol included several measures designed to mitigate such effects, complete elimination is not scientifically achievable due to their persistent and occasionally re-emergent nature across generations. Accordingly, this limitation is now clearly acknowledged in the revised version, consistent with your helpful suggestion (Discussion section, lines 436-445).
|
||
Comments 2: Second, include significance testing for the fixed effects: population, location, age, initial weight, sex, and SNP genotype, as a supplementary file. |
||
Response 2: Significance tests for each of the following fixed effects are given as a supplemental file (supplemental Table S1): population, location, age, initial weight, sex, and SNP genotype. For the fixed effects examined in the MLM association model for weight and length attributes across SNPs and InDels variants, the F-values, p-values, and R2 are provided in supplementary Table S1 (Heading 3.4.1, lines 351-354 and 668-669)
Comments 3: Third, regarding qPCR: we were not questioning the use of β-actin as the internal reference gene. The question was whether an additional reference gene could be included to improve data normality and accuracy. If that is not possible, this limitation should also be discussed in the revised manuscript. Response 3: We sincerely apologize for the earlier misunderstanding and are grateful to the reviewer for the clarification. We fully acknowledge that the use of multiple reference genes is widely recognized as a more robust and accurate approach for normalization in quantitative gene expression analysis. However, due to financial constraints, it was not feasible to include additional reference genes in the current study. Consequently, we relied on the β-actin gene, which has been previously validated in Oreochromis niloticus and shown to be a stable reference under comparable experimental conditions. In response to the reviewer’s suggestion, we have now explicitly acknowledged this as a limitation in the revised manuscript and indicated that incorporating multiple validated reference genes will be a priority in future work to further strengthen data reliability (Discussion section, lines 567–575). Comment 4: Finally, it is not good practice to cite studies with poor experimental design, especially as many of them have not undergone rigorous peer review. Response 4: We completely agree that citing research with robust experimental design and rigorous peer review is essential to uphold the scientific integrity of any publication. In response to your valuable comment, we have carefully revised and improved the reference list to ensure that only high-quality, relevant, and peer-reviewed studies are included. Additionally, we have updated the reference formatting to include the full journal names instead of abbreviations, in alignment with standard scientific citation practices. Furthermore, DOI addresses have been provided for all references wherever available, with the exception of seven references that do not have an associated DOI. We appreciate your emphasis on this critical issue, and this valuable guidance will be carefully considered and applied in all of our future work to further strengthen the quality and credibility of our scientific contributions. |